METHODS AND RESOURCES

# The structural context of posttranslational modifications at a proteome-wide scale

**Isabell Bludau[1], Sander Willems[1], Wen-Feng Zeng[1], Maximilian T. Strauss[2], Fynn M. Hansen[1], Maria C. Tanzer[1], Ozge Karayel[1], Brenda A. Schulman[3], Matthias Mann[1,2]***

**1** Department of Proteomics and Signal Transduction, Max Planck Institute of Biochemistry, Martinsried, Germany, **2** Proteomics Program, NNF Center for Protein Research, Faculty of Health Sciences, University of Copenhagen, Copenhagen, Denmark, **3** Department of Molecular Machines and Signaling, Max Planck Institute of Biochemistry, Martinsried, Germany

\* mmann@biochem.mpg.de

**Data Availability Statement:** All relevant data are within the paper and its Supporting Information files. Source data for the figures can be found at the Github repository (https://github.com/MannLabs/structuremap_analysis).

## Abstract

The recent revolution in computational protein structure prediction provides folding models for entire proteomes, which can now be integrated with large-scale experimental data. Mass spectrometry (MS)-based proteomics has identified and quantified tens of thousands of posttranslational modifications (PTMs), most of them of uncertain functional relevance. In this study, we determine the structural context of these PTMs and investigate how this information can be leveraged to pinpoint potential regulatory sites. Our analysis uncovers global patterns of PTM occurrence across folded and intrinsically disordered regions. We found that this information can help to distinguish regulatory PTMs from those marking improperly folded proteins. Interestingly, the human proteome contains thousands of proteins that have large folded domains linked by short, disordered regions that are strongly enriched in regulatory phosphosites. These include well-known kinase activation loops that induce protein conformational changes upon phosphorylation. This regulatory mechanism appears to be widespread in kinases but also occurs in other protein families such as solute carriers. It is not limited to phosphorylation but includes ubiquitination and acetylation sites as well. Furthermore, we performed three-dimensional proximity analysis, which revealed examples of spatial coregulation of different PTM types and potential PTM crosstalk. To enable the community to build upon these first analyses, we provide tools for 3D visualization of proteomics data and PTMs as well as python libraries for data accession and processing.

## Introduction

Posttranslational modifications (PTMs) are an important mechanism to regulate the activity and function of proteins. Mass spectrometry (MS)-based proteomics has become the method of choice not only to identify and quantify proteomes [1,2], but also to investigate PTMs on a proteome-wide scale [3–5]. Despite impressive technological progress, a key challenge in the PTM and signaling fields remains to distinguish PTMs that are of direct functional relevance from the tens of thousands that are routinely measured. This is necessary to match the proteomics data to dedicated, low-throughput biochemical follow-up studies that characterize the biological functions of candidate PTMs.

**Funding:** IB, SW, WFZ, MTS, FMH, MCT, OK, BAS and MM were supported by the Max-Planck Society for Advancement of Science. IB was supported by a Postdoc.Mobility fellowship granted by the Swiss National Science Foundation [P400PB_191046]. MM was supported by the Bavarian State Ministry of Health and Care through the research project DigiMed Bayern (www. digimed-bayern.de). The funders had no role in study design, data collection and analysis, decision to publish, or preparation of the manuscript.

**Competing interests:** The authors have declared that no competing interests exist.

**Abbreviations:** AKR1B1, aldo-keto reductase family 1 member B1; HSE, half-sphere exposure; IDR, intrinsically disordered region; MAPK3, mitogen-activated protein kinase 3; MAP4K1, mitogen-activated protein kinase kinase kinase kinase 1; MS, mass spectrometry; PAE, predicted aligned error; PDHA1, pyruvate dehydrogenase E1 component subunit alpha; pPSE, prediction-aware part-sphere exposure; PTM, posttranslational modification; RIPK2, receptor-interacting serine/ threonine-protein kinase 2; SASA, solvent accessible surface area.

To assess functional relevance of PTMs on a more global scale, Beltrao and colleagues recently presented a machine learning model that uses information from different features indicative of proteomic, structural, regulatory, or evolutionary relevance to derive a functional score for a large catalog of phosphosites [6]. Another recent study directly evaluated the functional relevance of phosphorylations purely based on available structural information [7]. Based on these and many previous studies, we know that phosphorylations are predominantly observed on spatially accessible amino acids and particularly in intrinsically disordered regions (IDRs) [7–9]. Furthermore, it stands to reason that phosphorylations in flexible regions within folded domains and on binding interfaces are more likely to be functional compared to those that are buried or less accessible in rigidly folded regions [7]. Although these studies impressively highlight the value of integrating structural information into the analysis of PTMs, they have been limited to phosphorylation and to the set of available experimentally derived structures deposited in PDB, which furthermore inherently favor stable regions of proteins [10].

Recently, there has been a key breakthrough in computational protein structure prediction from just the amino acid sequence of a protein. The novel deep learning models in AlphaFold2 (henceforth referred to as AlphaFold) [11], rapidly followed by RoseTTAFold [12], were shown to regularly achieve high accuracy in predicting protein structures that are largely comparable to those determined by experimental methods. By providing structural information for almost the complete human proteome as well as the proteomes of over 20 model organisms, the AlphaFold protein structure database (AlphaFold DB; https://alphafold.ebi.ac.uk) now enables structural investigations on a proteome-wide scale, thus promising to accelerate our understanding of the structure-to-function relationship of proteins [13,14]. It has further been proposed that AlphaFold has immense potential for incorporating and analyzing PTMs, for instance, glycosylation [15].

Here, we set out to systematically combine the wealth of structural information from AlphaFold with proteomics data, especially large-scale PTM information, with the goal of shedding new light on the long-standing question of functional relevance of PTMs. We present a first systematic assessment of how PTM data can be integrated with deep learning–predicted structures on a proteome-wide scale. We then explore key features in the structure function domain by combining predictions of functional relevance with domain features and discover a multitude of sites with potential regulatory roles. To enable the community to further explore the numerous related biological questions, we provide a Python package called StructureMap, which allows to easily and quickly access and integrate structural data from AlphaFold DB with proteomics data and information on PTMs. Finally, we provide an extended version of our previously published AlphaMap tool for sequence visualization [16], which now enables the mapping of peptides and PTMs to three-dimensional protein structures.

## Results

### Estimation of side chain exposure and intrinsically disordered regions from predicted protein structures

To make the information provided by predicted structures accessible for systematic analyses, we first wanted to extract it into tractable metrics such as amino acid side chain exposure or the categorization of amino acids into structured and intrinsically disordered regions (IDRs). At this point, it is important to make a clear distinction between predicted and experimentally derived structures. Experimental structures are often incomplete and may only cover a specific sequence region. In contrast, the predicted structures in AlphaFold DB in principle cover the entire protein sequence from N- to C-terminus. Importantly, each amino acid in the predicted

structure is associated with a specific prediction confidence (pLDDT) derived from the deep learning models [11]. Additionally, the relative position of amino acids to each other is annotated with an expected distance error in Ångströms (predicted aligned error (PAE)). Although experimental structures are available for a large set of proteins, we decided to base all our analyses on predicted structures only. This prevents ambiguities in the integration of multiple conflicting structures and allows us to leverage the complete sequence information, confidence metrics, and PAE estimates.

We found that it is crucial for all structure-based calculations to take AlphaFold's confidence and PAE metrics into account for best accuracy, as also pointed out before [17]. This ensures that the predicted structures from AlphaFold are not considered as "static," but as estimates with a certain degree of positional uncertainty that varies across the protein sequence. This is particularly important for estimating metrics for intrinsically disordered proteins and proteins with longer IDRs. Therefore, we developed a prediction-aware metric to evaluate amino acid side chain exposure. For this, we built upon the previously introduced half-sphere exposure (HSE) [18,19]. This method essentially calculates the half-sphere of a given amino acid in the direction of its side chain at a defined radius and counts the number of alpha carbon atoms from other residues within it, with a larger number reflecting less exposure and vice versa (S1A Fig). We adjusted the HSE to take prediction uncertainties into account, meaning that an alpha carbon atom is only considered as a neighbor if it still lies within the defined radius after addition of its PAE for this alpha carbon pair. We further introduce an angle parameter that determines whether to consider the full-sphere, half-sphere, or any other angle in direction of the amino acid side chain. Accordingly, we termed our metric prediction-aware part-sphere exposure (pPSE). To illustrate, Fig 1A shows the AlphaFold predicted structure of mitogen-activated protein kinase 3 (MAPK3) colored by pLDDT and by the pPSE using a radius of 12 Å and an angle of 70˚. We chose these values after considering the average size of amino acids of approximately 3.5 Å and side chain flexibility around the direction of the beta carbon (please refer to the Methods section for further details).

The higher the pPSE, the more other amino acids are in close proximity to the amino acid being evaluated and, hence, the more structured its environment. In this respect, pPSE offers a similar metric as the commonly used solvent accessible surface area (SASA), or relative SASA (RSA). However, the pPSE directly considers side chain orientation, and, more importantly, it takes the prediction error of AlphaFold into account. Estimating the pPSEs for all amino acids in the 20,053 predicted human protein structures on AlphaFold only takes minutes on a laptop computer with our implementation. This makes the tool especially useful for system-wide studies where tens of thousands of proteins are evaluated for particular properties. In a community effort, it was recently shown that a smoothed AlphaFold confidence metric (pLDDT) or RSA metric based on the predicted structures can confidently determine IDRs [17], improving on IUPred2, a state-of-the-art tool for IDR prediction [20]. We found that our smoothed pPSE metric even obtains slightly better results when using a radius of 24 Å and a full sphere (the true positive rate improves from 83% for RSA to 86% for pPSE; see Fig 1A, 1B, and 1C and S1C and S1D Fig). Importantly, considering the positional uncertainty during pPSE estimation considerably improves IDR prediction compared to neglecting it (TPR increase from 79% to 86%).

## Most PTMs are enriched in intrinsically disordered regions, whereas ubiquitinations accumulate in structured domains

Having the proteome-wide information on IDRs at hand, we next performed an enrichment analysis of different PTMs located within those regions across the entire human proteome. We

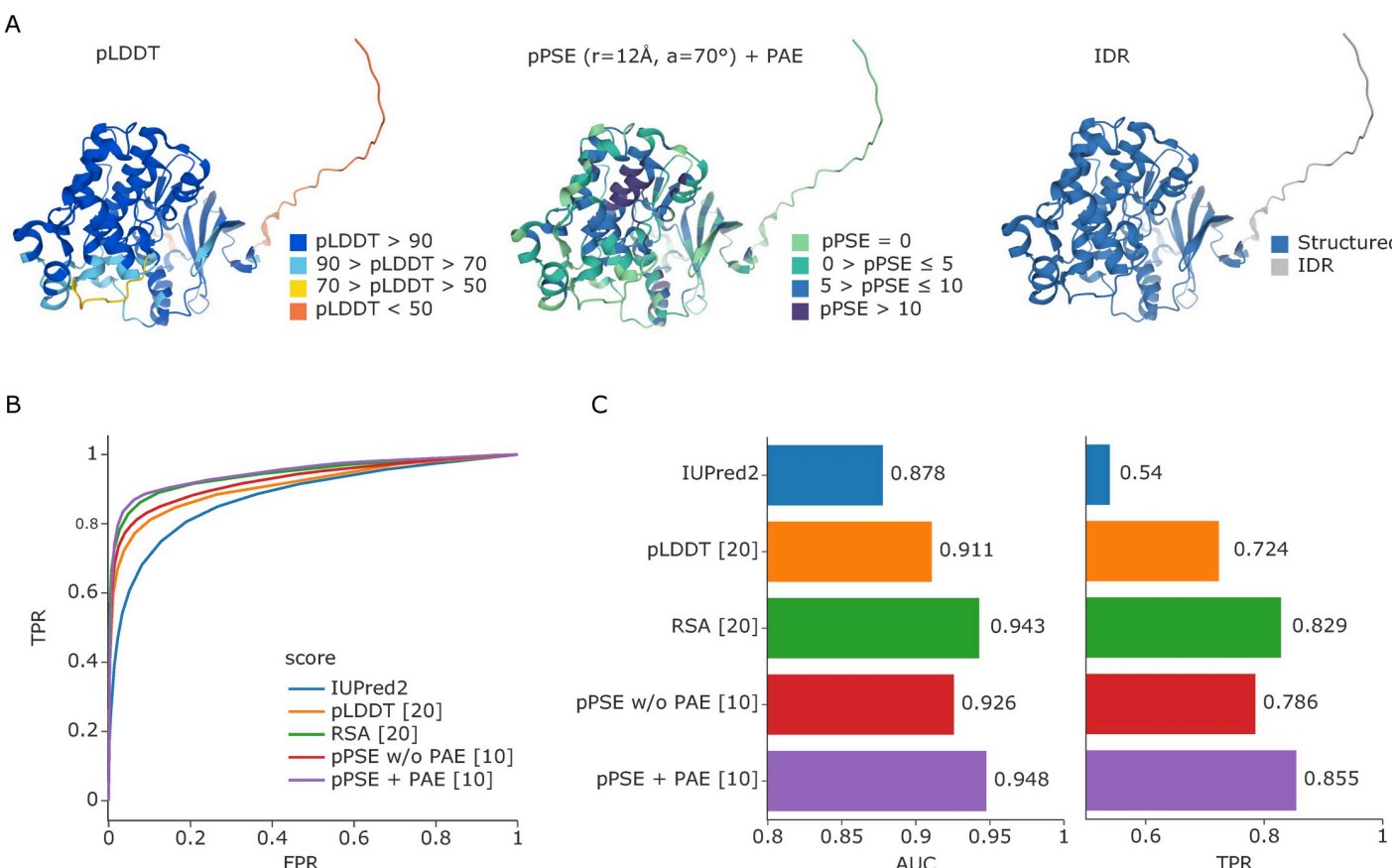

**Fig 1. Estimation of amino acid side chain exposure and IDRs. (A)** AlphaFold predicted structure of MAPK3 colored by prediction confidence (pLDDT, left), colored by our pPSE metric using a radius of 12 Å and an angle of 70˚ (center), and colored by our prediction of structured regions and IDRs (right). **(B)** ROC curve for predicting IDRs based on IUPred2 in comparison to the smoothed pLDDT confidence scores from AlphaFold, the smoothed RSAs, and the pPSE with (+) and without (w/o) considering the PAE (radius = 24 Å, angle = 70˚). **(C)** Corresponding AUC values and the TPRs at a 5% FPR. The numbers in square brackets behind each metric indicate the smoothing windows that were used; see S1 Fig for a comprehensive parameter screen. Source data for (B) and (C) are available at Github. AUC, area under the curve; FPR, false positive rate; IDR, intrinsically disordered region; MAPK3, mitogen-activated protein kinase 3; PAE, predicted aligned error; pPSE, prediction-aware part-sphere exposure; ROC, receiver operating characteristice; RSA, relative solvent accessible surface area; TPR, true positive rate.

first used all PTMs annotated in the PhosphoSitePlus database that overlapped with structural data, comprising a total of 334,529 sites, including phosphorylations (p), ubiquitinations (ub), sumoylations (sm), acetylations (ac), methylations (m), and the glycosylations O-GlcNAc (gl) and O-GalNAc (ga) [21]. In agreement with previous observations of phosphorylations [8,9], most PTMs were indeed significantly enriched in IDRs (Fig 2A). In contrast, our analysis revealed that ubiquitinations and, to a lesser extent, acetylations were significantly underrepresented in IDRs (ubiquitination: odds-ratio = 0.6, adj. $p$-value $\approx 0$, number of sites = 91,388; acetylation: odds ratio 0.9, adj. $p$-value = $6 \times 10^{-8}$, number of sites = 21,202). However, if only PTM sites with a known regulatory function are considered, this effect disappeared for ubiquitination and was even reversed for acetylation (ubiquitination: odds ratio = 1.0, adj. $p$-value = 0.7, number of sites = 451; acetylation: odds ratio 2.0, adj. $p$-value = $1 \times 10^{-18}$, number of sites = 631).

A possible explanation for nonregulatory ubiquitination sites in structured regions is the tagging of misfolded proteins for degradation by the proteasome. Importantly, most datasets that contribute ubiquitination sites to PhosphoSitePlus are from samples treated with proteasome inhibitors (Fig 2B). This leads to the accumulation of misfolded proteins that presumably

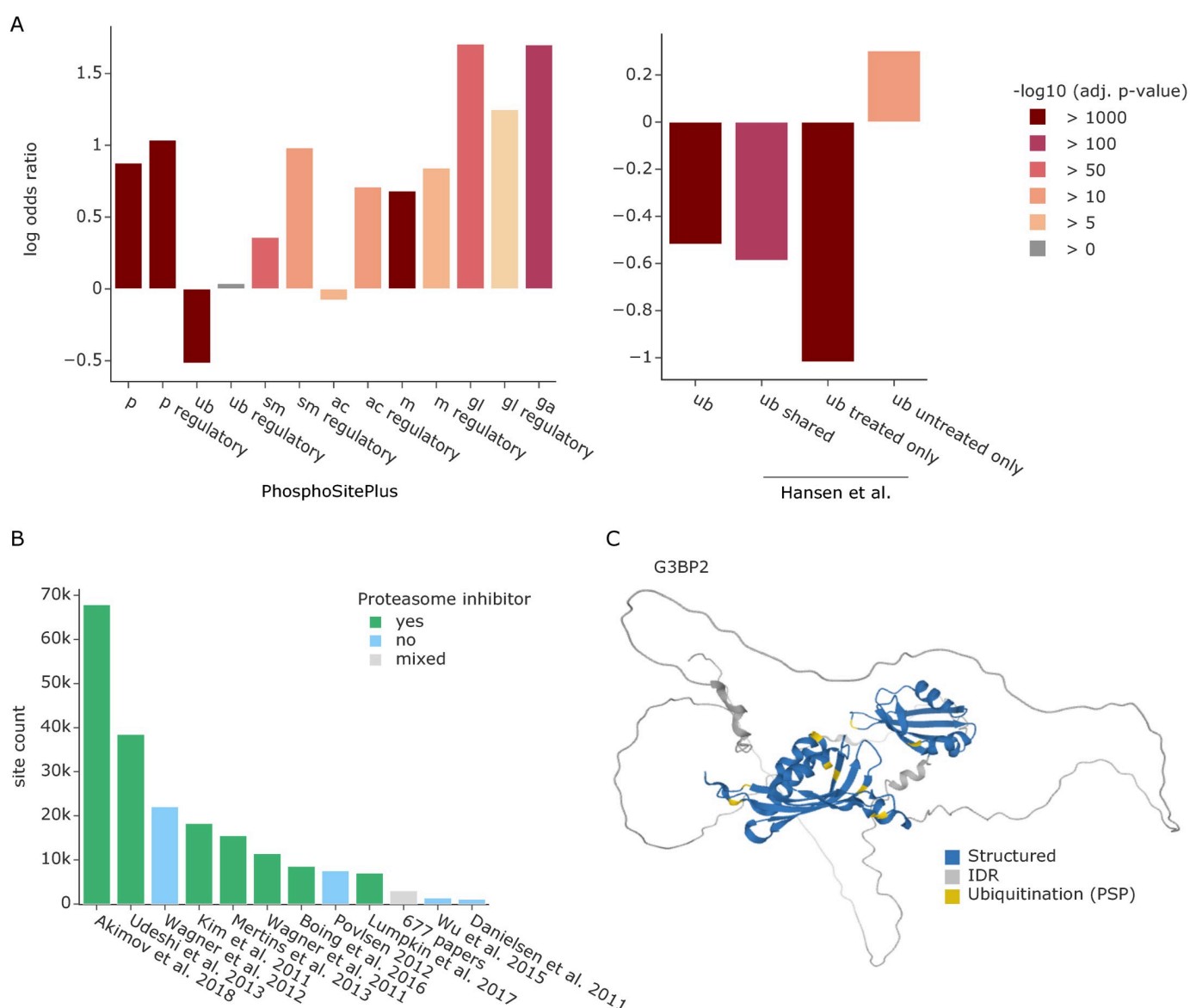

**Fig 2. Enrichment analysis of PTMs in IDRs. (A)** Enrichment of different PTMs annotated in the PhosphoSitePlus database in IDRs (left). Enrichment of ubiquitinated lysines annotated in PhosphoSitePlus versus ubiquitinations detected in a dataset treated with proteasome inhibitor or untreated (right) [4]. PTMs are abbreviated as follows: phosphorylations (p), ubiquitinations (ub), sumoylations (sm), acetylations (ac), methylations (m), and the glycosylations O-GlcNAc (gl) and O-GalNAc (ga). Source data are available at **Github**. **(B)** Overview of datasets that contribute ubiquitination sites to PhosphoSitePlus. Publications reporting >1,000 ubiquitination sites are listed separately and are colored based on their use of proteasome inhibitors. The additional 677 studies that contribute fewer ubiquitination sites were aggregated to a gray bar irrespective of their use of proteasome inhibitors. Source data are available at **Github**. **(C)** AlphaFold predicted structure of Ras GTPase-activating protein-binding protein 2 (G3BP2) colored by structured regions (blue) and predicted IDRs (gray) as well as ubiquitination sites annotated in PhosphoSitePlus (yellow). IDR, intrinsically disordered region; PTM, posttranslational modification.

expose normally inaccessible lysine residues. Furthermore, ubiquitin might also be specifically attached to structured regions to destabilize the protein fold, creating new short IDRs that are required for proteasome binding and subsequent degradation [22].

To directly test our hypothesis, we contrasted ubiquitination sites from proteasome inhibitor-treated and untreated samples in the same experiment [4]. Interestingly, ubiquitination sites unique to the proteasome inhibition condition confirm the overrepresentation of ubiquitination in structured regions (Fig 2A right, odds ratio = 0.4, adj. *p*-value = 0, number of

sites = 19,517). The same effect can still be observed for the sites shared between both datasets (odds ratio = 0.6, adj. $p$-value = $1 \times 10^{-193}$, number of sites = 11,741), whereas the ubiquitination sites unique to the untreated condition are enriched in IDRs (odds ratio = 1.4, adj. $p$-value = $1 \times 10^{-32}$, number of sites = 6,321), similar to most other PTMs. Overall, 78% of ubiquitinated lysines unique to the inhibitor treatment condition and 71% of shared ubiquitin sites were in structured regions. In notable contrast, in the uninhibited condition, only 50% of observed ubiquitinations are in structured regions.

To further pursue these findings, we disregarded all amino acids in predicted IDRs and asked if PTMs are enriched in amino acids with side chains of high versus low exposure within structured regions. To this end, we calculated the pPSE for each amino acid at a radius of 12 Å and an angle of 70˚. We considered amino acids with a pPSE $\leq 5$ to have a high exposure and those with a pPSE $> 5$ as low exposure (see S1B Fig and the Methods section for details on the cutoff selection). Due to much lower numbers of annotated PTMs in structured regions, statistical significance decreases, but phosphorylations were still enriched in amino acids with high side chain exposure, whereas ubiquitinations were enriched in those with low side chain exposure (S2 Fig). This indicates that ubiquitinations are located on lysines that are buried within the structure of a properly folded protein rather than on outwards facing amino acids of a helix or beta-sheet at the protein's surface.

In addition to these global analyses, we further explored modified proteins individually to test if PTMs were specifically enriched in certain structural elements. For ubiquitination, this revealed 71 proteins with a significant enrichment in structured regions (odds ratio $< 1$, adj. $p$-value $\leq 0.05$). Interestingly 80% of them were DNA or RNA binding proteins. A striking example is Ras GTPase-activating protein-binding protein 2 (G3BP2), an RNA-binding protein that plays an essential role in cytoplasmic stress granule formation [23]. Here, all 9 ubiquitination sites are in structured regions (Fig 2C). Interestingly, 6 of those ubiquitinations are localized in the NTF2-like domain, which ranges from amino acids 11 to 133 and plays an important role in G3BP2 oligomerization and the binding of deubiquitinating enzyme 10 (USP10) [23].

## Improving sequence motif analysis through structural context

PTMs are commonly introduced by dedicated enzymes such as kinases for phosphorylation, E3-ligases for ubiquitination, and proteases for proteolytic cleavage, which generally recognize specific sequence motifs. Given that most PTMs have a preference for exposed amino acids, we reasoned that sequence motifs in accessible protein regions should be preferred compared to inaccessible ones, adding another layer of selectivity. To explore this hypothesis, we first selected a curated list of kinase phosphorylation motifs available in Perseus [24]. Based on phosphosites in both PhosphoSitePlus and a recent in-depth, COVID-related phospho study [25], we first confirmed that phosphorylations are generally enriched in kinase phosphorylation motifs compared to all serines, threonines, and tyrosines (STY sites) in the proteome. This effect is even more pronounced for regulatory sites and sites from the Stukalov study (Fig 3A). Confirming our hypothesis, motifs in IDRs and motifs harboring exposed STY sites were indeed preferentially modified (Fig 3B and S3A Fig).

These results highlight that proteome-wide structural information can provide valuable insights for motif analysis and help interpret experiments determining enzyme-substrate relationships. We illustrate this with a large in vitro kinase substrate screen by Ishihama and colleagues [26]. These authors dephosphorylated HeLa cell lysates with phosphatases, which they then deactivated by heat. The resulting—partially denatured—sample then served as a substrate pool to which 385 different recombinant human protein kinases were individually

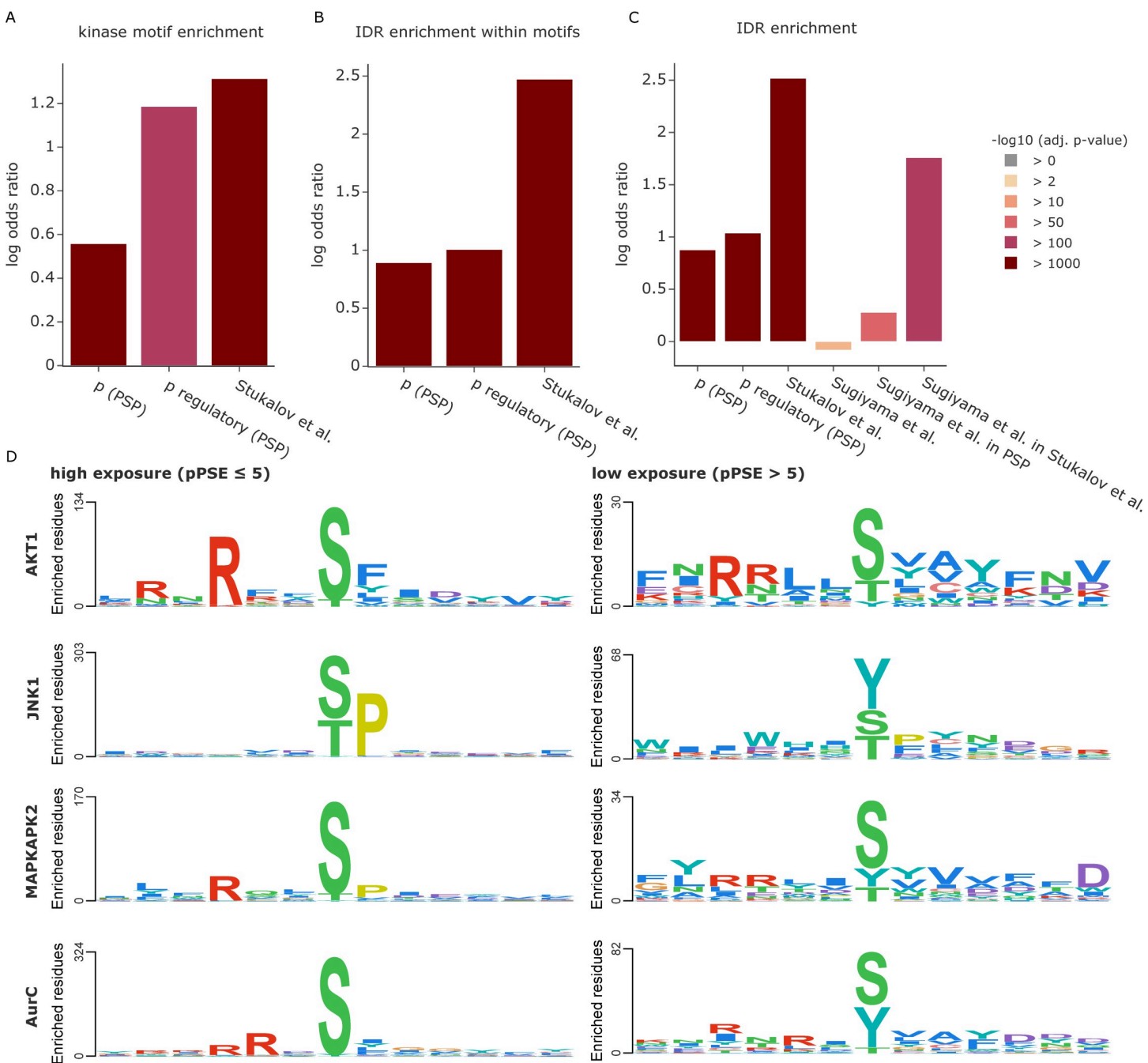

**Fig 3. Exploiting the 3D context of kinase phosphorylation motifs. (A)** Enrichment of phosphorylation events in kinase motifs compared to all possible STY sites. PSP stands for PhosphoSitePlus. Source data are available at Github. **(B)** Enrichment of phosphorylations in kinase motifs within IDRs compared to all possible kinase motif occurrences. Source data are available at Github. **(C)** Enrichment of phosphorylations in IDRs compared to all possible STY sites. The phosphosites reported by Sugiyama and colleagues [26] were filtered for sites also reported in PSP or by Stukalov and colleagues [25]. Source data are available at Github. **(D)** Sequence logos for different kinases in context of the dataset from Sugiyama and colleagues [26]. Motifs for phosphosites of high exposure (pPSE ≤ 5) are shown on the left and phosphosites of low exposure (pPSE > 5) are shown on the right. The PSSMSearch tool [27] was used with a log odds scoring method [28]. IDR, intrinsically disordered region; pPSE, prediction-aware part-sphere exposure; PSP, PhosphoSitePlus.

added to investigate which kinases phosphorylate which specific amino acid sites. Their study resulted in an unprecedented set of 20,669 phosphosites, 175,574 proposed kinase-substrate relationships, and 1,427 kinase phosphorylation motifs.

Based on the denaturation of the proteome, and as already indicated by the authors, we speculated that many of the previously inaccessible sites would now be amenable to phosphorylation, providing an ideal test case for structure-based interpretation. Indeed, our analysis revealed that the identified phosphosites were not enriched in IDRs, in contrast to the above examples. However, the sites that overlapped with other studies did show the expected enrichment in IDRs (Fig 3C). These results suggest that structural information can be used to refine the list of reported phosphosites to a set that better represents the sites expected to occur on endogenously folded proteins.

To further test the effect of 3D exposure filtering on sequence motif analysis in context of the dataset from Sugiyama and colleagues [26], we selected kinases and performed a motif analysis separately for sites of high surface exposure (pPSE ≤ 5) and sites of low exposure (pPSE > 5) using the PSSMSearch tool [27,28]. As can clearly be seen in Fig 3D, kinases showed striking differences in sequence motifs between sites of high and low exposure. While motifs for sites of high exposure are mostly in agreement with the reference set provided by Perseus [24], this was not the case for the sites with low exposure. To account for the fact that there are fewer sites in structured regions, we also selected random, equally sized subsets of the high-exposure sites and repeated the motif analysis. This resulted in similar patterns as for the full set of sites, but with lower enrichment scores (S3B Fig).

In the case of the RAC-alpha serine/threonine-protein kinase (ACT1), the phosphosites of high exposure clearly display the R-x-R-x-x-pS-F motif (Fig 3D, top left panel). In contrast, phosphosites of low exposure only provide a noisy motif (Fig 3D, top right panel). For other kinases, such as stress-activated protein kinase JNK1, a serine/threonine-specific protein kinase, the phosphosites of low exposure even have an unexpected enrichment for a phosphorylated central tyrosine residue and the proline at the +1 position is hardly enriched (Fig 3D, right panel).

Together, these results establish that the structural information from AlphaFold and the tools presented herein can guide determining potential regulatory PTM sites found by in vitro screens, increasing the confidence of measured kinase–substrate pairs by filtering out a subset that are less likely to be true in vivo substrates. As we have shown, this can improve kinase phosphorylation motif predictions and help to interpret individual sites of interest. Here, we focused on phosphorylations but we expect similar benefits for the analysis of any other types of motifs, including enzyme recognition or general protein binding.

### Functionally relevant PTMs are enriched in short IDRs within large structured domains exemplified by kinase activation loops

Having established the enrichment of most PTMs in disordered regions in the dataset, we further explored the structural context of this effect. We found that these PTMs are often located in short IDRs that are embedded within larger structured domains. To investigate if this was a random effect or whether these short IDRs could be of biological relevance, we extracted all proteins with short IDRs of maximally 20 amino acids length between 2 flanking structured regions of at least 80 amino acids. Among the 20,053 human proteins in AlphaFold DB, 2,454 have such a pattern. Notably, enrichment analysis of these proteins revealed a significant over-representation of GO molecular functions related to ATP binding, protein kinase activity, ATPase activity, transmembrane transporter activity, and motor activity (Fig 4A).

To further evaluate the relevance of the short IDRs, we analyzed their occupancy with functional phosphosites. To this end, we first considered regulatory phosphosites in PhosphoSite-Plus [21]. Our analysis revealed that these regulatory sites, as compared to phosphosites in general, are significantly enriched in short IDRs versus all IDRs (odds ratio = 1.57, adj. p-value = 0.001; Fig 2B). We also extracted a second set of phosphosites from the abovementioned

study of Beltrao and colleagues, where sites were given a functional score between 0 and 1, with scores equal to or above 0.5 considered functional [6]. We indeed observed that the higher the score, the stronger the enrichment of functional phosphosites in short IDRs, ranging from an odds ratio of 1.43 at a cutoff of 0.5 to an odds ratio of 7.7 at 0.9 (Fig 4B). This raises the exciting possibility that at least a subset of these phosphorylation sites may play important roles in structural and functional rearrangements of their neighboring domains.

Due to the strong enrichment of kinases among proteins with short IDRs, we next investigated whether they overlap with any known kinase substructures annotated in KinaseMD [29]. That database contains substructure annotations for 388 kinases, 365 of which also have predicted structures in AlphaFold DB. We found a large overlap of 72 short IDRs with the 309 annotated activation loops, but none with the 171 G-loops or 230 Cα-helix positions in these kinases. If the 5 amino acids flanking a short IDR are also considered, this number increases to 79 (also see Methods section). These results are particularly interesting, because the activation loops of many kinases undergo structural rearrangements upon phosphorylation [30]. Strikingly, 55 of the 79 kinases (70%) with an overlap of the extended short IDR and the annotated activation loop have an annotated regulatory phosphosite in PhosphoSitePlus or a functional score higher than 0.5 (as determined by Beltrao and colleagues [6]), for a total of 99 different phosphosites. To illustrate, receptor-interacting serine/threonine-protein kinase 2 (RIPK2) and mitogen-activated protein kinase kinase kinase kinase 1 (MAP4K1) both show an overlap between our predicted short IDR and the annotated activation loop with known regulatory phosphosites (Fig 4C, 4D, and 4E).

Next, we evaluated short IDRs outside of annotated kinase activation loops. An interesting example of these is the serine/threonine-protein kinase CHK2 (CHEK2). Although the annotated activation loop (amino acids 367 to 389) was not detected as a short IDR, our data contained an alternative short IDR region (amino acids 262 and 263). Directly flanking this IDR is a phosphorylation site with a functional score of 0.44 (S260). Notably, CHEK2 was reported to be autophosphorylated at residue S260, which is important for triggering a conformational change in CHEK2 that favors a dissociation of dimers into fully active monomers [31].

We also observed short IDRs in many proteins apart from kinases. One example is Band 3 anion transport protein (SLC4A1), where we found that the short IDR from residue 354 to 369 contains a known regulatory phosphosite on Y359 [32]. In addition to phosphorylations, other PTMs might also be biologically relevant in the short IDRs. Indeed, 1 of 3 regulatory ubiquitination sites in another solute carrier protein—SLC22A6—is located directly in a short IDR, whereas the other 2 are in close proximity (Fig 4F). These 3 ubiquitination sites have previously been shown to play an important role for the internalization of this protein [33].

Compared to phosphorylation, the percentage of other PTM sites with known functions is even smaller. Our findings suggest that selecting candidates from PTM sites within or in close proximity to short IDRs is a promising strategy to discover functional relevance. We found that our predicted short IDRs extended by 5 amino acids contain a wealth of PTM sites that are not yet annotated as regulatory in PhosphoSitePlus (1,437 phosphosites, 898 ubiquitination sites, 118 acetylations, 43 sumoylations, 53 methylations, 33 GalNAc, and 1 GlcNAc) (Fig 4G, S1 Dataset). We further provide a list of all human short IDRs for researchers to explore their favorite proteins, enabling the integration of own experimental data from PTM studies or other types of studies, such as mutational screens (S2 Dataset).

## PTMs on proteins preferentially occur in three-dimensional clusters

It is well known that many proteins have hotspots of modifications. For example, multisite phosphorylation in specific sequence regions is critical in regulating the activity of many

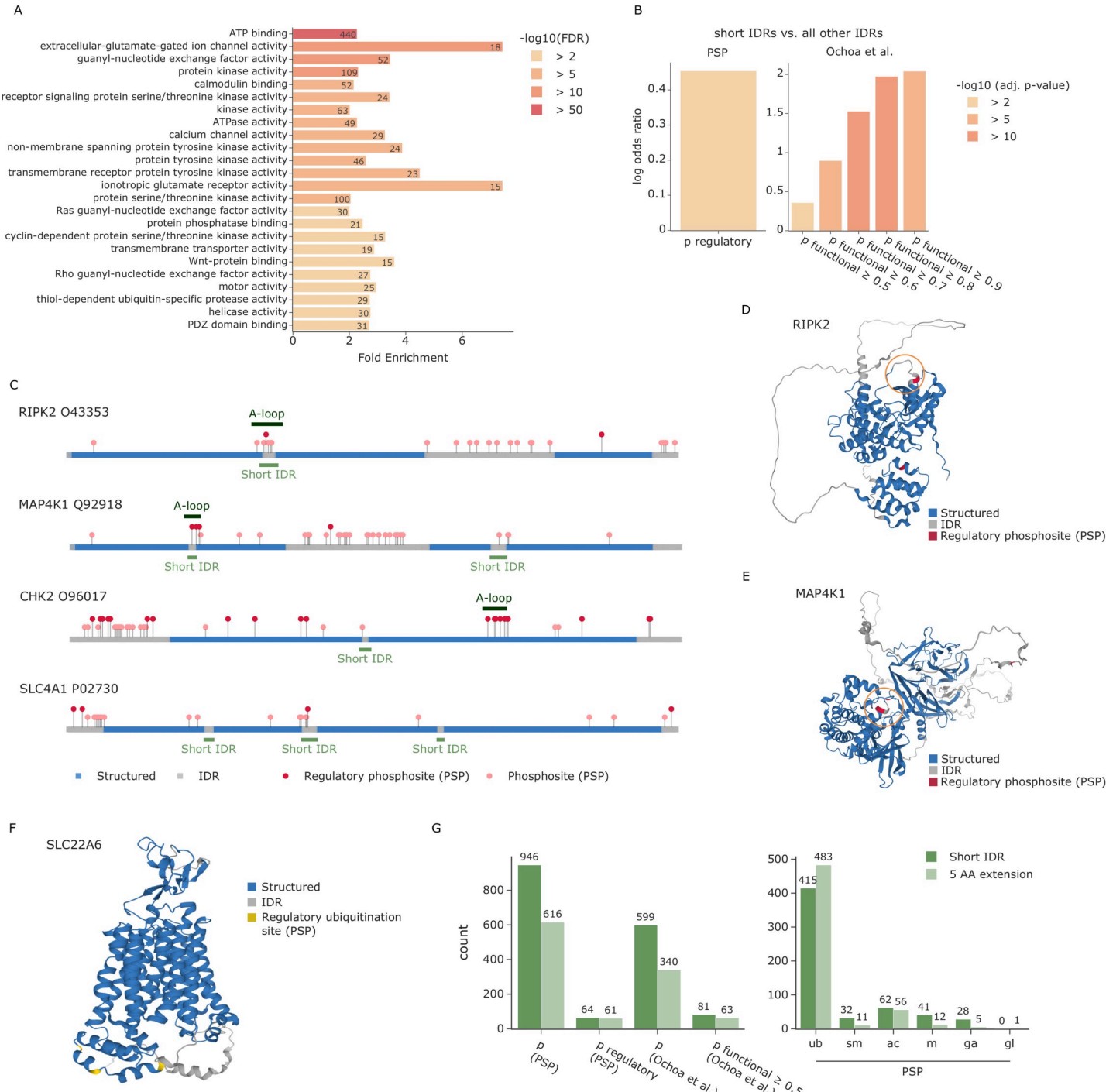

**Fig 4. Regulatory PTMs accumulate in short IDRs.** **(A)** Enrichment analysis of proteins with short IDRs. Source data are available at Github. **(B)** Enrichment of regulatory phosphosites from PSP in short IDRs compared to all other IDRs. Source data are available at Github and Github, respectively. **(C)** Sequence plot showing the N- to C-terminus of different proteins colored by whether the amino acid is part of a structured region (blue) or an IDR (gray). All phosphosites annotated in PSP are indicated by circles. Regulatory sites are colored in dark red and stand out higher than nonregulatory sites (salmon). Regions of short IDRs including a 5-amino acid extension are indicated in light green below the sequence. Annotated kinase activation loops (A-loops) from KinaseMD are indicated in dark green above the sequence. The predicted structures of RIPK2 **(D)** and MAP4K1 **(E)** are colored by structured regions (blue) and predicted IDRs (gray) as well as known regulatory phosphosites annotated in PSP (dark red). Specific regions of interest are highlighted by an orange circle. **(F)** The predicted structure of SLC22A6 is colored by structured regions (blue) and predicted IDRs (gray) as well as known regulatory ubiquitination sites annotated in PSP (yellow). **(G)** Overview of phosphorylations (left) and other PTMs (right) that lie within short disordered regions or their flanking 5 AAs. Source data are available at Github. AA, amino acid; IDR, intrinsically disordered region; MAP4K1, mitogen-activated protein kinase kinase kinase kinase 1; PSP, PhosphoSitePlus; PTM, posttranslational modification; RIPK2, receptor-interacting serine/threonine-protein kinase 2.

enzymes and their binding properties [34]. Furthermore, in the context of circadian biology, we recently showed that regulated ubiquitinations frequently occur in sequence proximity [4]. Beyond colocalization of the same PTM types, phospho-acceptors near PTM-modified lysines were shown to be preferentially phosphorylated in comparison to more distant residues [35]. Those prior findings were obtained on the basis of linear sequence analysis. Now, with the spatial coordinates of each PTM acceptor residue provided by AlphaFold, we set out to evaluate PTM proximity in three-dimensional space.

First, we investigated whether PTM acceptors near a modified amino acid residue are more frequently observed to also be modified compared to more distant residues or to random expectation. For this, we extended the strategy of Krogan and colleagues [35] to evaluate distance in 3D space and to assess both individual PTM types and PTM co-occurrence (see Methods section). Importantly, our metric considers the predicted positional uncertainty between any 2 PTM sites as a factor in the analysis. This ensures that uncertainties in the relative positioning of folded domains that are linked by flexible IDRs are considered in the proximity calculation. Furthermore, we only take structured regions and short IDRs into account for the proximity analysis (that is, we removed all IDRs of more than 20 amino acids). This ensures that proximity results are not influenced by regions of high structural uncertainty, as is the case for IDRs. It further avoids any biases that arise from the fact that many PTMs are enriched in IDRs and tend to cluster there in linear sequence space.

Our analysis revealed that the observed sites of PTM types annotated in PhosphoSitePlus indeed form 3D modification hotspots. Phosphorylations, ubiquitinations, sumoylations, acetylations, and methylations each form tight clusters in 3D space, where proximal amino acids are preferentially modified compared to more distant residues or an equivalent random selection (Fig 5A). Due to an overall lower number of O-GlcNAc and O-GalNAc modified sites, the results for these modifications are less conclusive but also show a similar trend (S4 Fig). In addition to evaluating PTM types by themselves, we further investigated colocalization of different PTM types. This confirmed that phospho-acceptors near modified lysines (including ubiquitination, sumoylation, and acetylation) are more frequently phosphorylated compared to random expectation [35] (Fig 5B). This is also true for methylated lysines and arginines. Conversely, investigating ubiquitination sites near other PTMs revealed that they also preferentially occur close to phosphosites (Fig 5C). Other lysine modifications, however, often compete for the same or directly neighboring residues, but they do not generally favor proximity. Overall, our analysis reveals that many proteins have specific 3D regions and folds that are particularly prone to being modified by the same or different PTMs. This structurally supports the notion of PTM cross-talk.

Following these global analyses, we next explored 3D PTM clusters of all individual proteins. For this, we calculated all pairwise distances between modified amino acids and compared their average against a distribution of random PTM sites (see Methods section). We again only considered structured protein regions and included the positional uncertainty between any 2 PTM sites in the distance calculation. Clustering analysis of phosphorylation and ubiquitination sites in PhosphoSitePlus revealed many proteins with significant PTM clusters, showing a strong enrichment for transmembrane proteins (Fig 5D). On those proteins, we detected these PTMs on the cytosolic domains and in 3D proximity, nicely confirming that the proximity analysis worked as intended.

To enable a more fine-grained inspection of PTM clusters, we explored an in-house phospho-dataset [25]. Of 47 phosphoproteins with 3 or more sites in structured regions, 3D proximity analysis yielded 6 significant ones (adj. $p$-value $\leq 0.05$ and $\geq 3$ phosphosites; S3 Dataset). As an example, the mitochondrial pyruvate dehydrogenase E1 component subunit alpha (PDHA1) had an adjusted proximity $p$-value of 0.004, and all 6 detected phosphosites

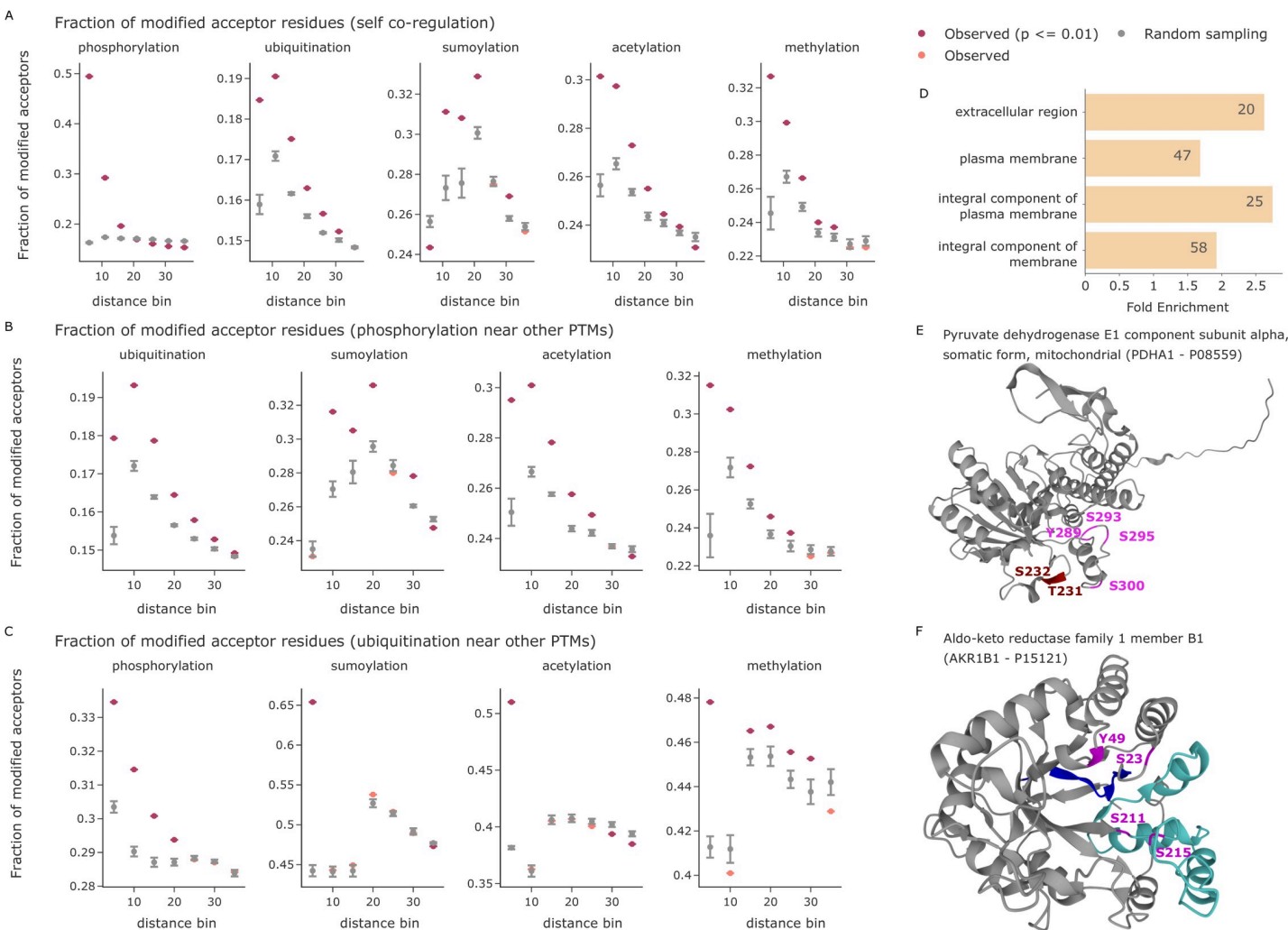

**Fig 5. PTM proximity analysis in 3D. (A)** The fraction of modified PTM acceptor residues is shown as a function of the 3D distance to a given modified amino acid in Å. Observed values (indicated in red when statistically significant and colored in salmon otherwise) are compared to the mean of 5 random samples including the same number of modified PTM sites (gray). Error bars indicate one standard deviation. The x-axes are divided in distance bins ranging from each previous bin to the indicated cutoff in Å. Source data are available at Github. **(B)** The fraction of modified phospho-acceptor residues is shown as function of the 3D distance to a given modified amino acid in Å. Source data are available at Github. **(C)** The fraction of ubiquitinated lysines is shown as function of the 3D distance to a given modified amino acid in Å. The smallest bin shows competition for the same central lysine residue. Source data are available at Github. **(D)** Enrichment analysis of proteins with 3D phospho- and/or ubiquitination clusters (FDR ≤ 0.01). Source data are available at Github. **(E)** The predicted structure of PDHA1. Phosphorylations on the phospho-loop A are indicated in dark red (T231 and S232). The phosphorylations on phospho-loop B are indicated in magenta (Y289, S293, S295, and S300) [36]. **(F)** The predicted structure of AKR1B1. Residues annotated as NADP binding sites are highlighted in blue (amino acids 10 to 19) and turquoise (amino acids 211 to 273). Phosphorylations are indicated in magenta. AKR1B1, aldo-keto reductase family 1 member B1; PDHA1, pyruvate dehydrogenase E1 component subunit alpha; PTM, posttranslational modification.

are located within 1 protein pocket (Fig 5E). Three of these (S232, S293, and S300) have previously been reported to be substrates of PDK family kinases. Any single one of these is sufficient to inactivate PDHA1, and dephosphorylation of all sites is required for reactivation [36]. Our 3D analysis suggests that the other 3 phosphorylation sites would have the same effect. Another interesting example is aldo-keto reductase family 1 member B1 (AKR1B1) (adj. *p*-value = 0.03). In linear sequence space, S211 and S215 are close together, whereas S23 and Y49 are far apart. However, in the folded protein structure, all 4 are in the same pocket (Fig 5F). Interestingly, it also contains the annotated NADP binding site of AKR1B1, which consists of 2 distinct sequence stretches from amino acids 10 to 19 and 211 to 273 (based on UniProt annotation).

## Community resources for enabling the systematic integration of PTM data with structure predictions from AlphaFold

This study only scratches the surface of biological insights that can be gained from combining PTM data with structural information. To enable the community to further investigate various research questions of interest to them in their own data or in public repositories, we created a toolset that facilitates systematic data exploration and integration. First, we provide Structure-Map, an open-source Python package for processing predicted structures from AlphaFold DB and for integrating the data with PTM information. Its functionalities include (1) accession of predicted structures from AlphaFold DB and extraction of essential information into an internal data format; (2) calculation of the pPSEs of individual amino acids as well es estimation of IDRs; (3) extraction of short IDRs for PTM site prioritization; (4) import and formatting of PTM datasets; (5) enrichment analyses of PTMs in different structural regions and IDRs; (6) motif analysis in 3D context and filtering based on side chain exposure; and (7) multidimensional proximity analysis of PTMs. To further enable easy visualization of PTMs on the three-dimensional structure of proteins, we also extended our previously published AlphaMap tool, which is available as Python library as well as a stand-alone application with graphical user interface [16]. The source code of both tools is openly available with an Apache license on the MannLabs GitHub page and includes extensive documentation to readily enable researchers to understand and further adopt code to any specific needs (https://github.com/MannLabs/ structuremap and https://github.com/MannLabs/alphamap). To support scientific transparency and reproducibility, we further provide data and python Jupyter notebooks to reproduce all analyses and figures presented herein (https://github.com/MannLabs/structuremap_ analysis).

## Discussion and outlook

PTMs provide essential mechanisms to regulate the activity and function of proteins. Although MS-based proteomics routinely enables the identification and quantification of thousands of PTMs, systematic assessment of their functional relevance remains a persisting challenge. While previous work already demonstrated the merits of structural information for PTM analyses [6–9,35], the recent revolution in computational protein structure prediction [11,12] only now enables the proteome-wide integration of structural information with PTM data. In this study, we provide a first overview of how the comprehensive structural context of all detected PTMs can provide global and protein-specific insights into biological mechanisms, to filter in vitro datasets for physiological relevance and to identify promising candidates for biochemical follow-up studies.

Key to most of our analyses was that we used whole proteome structural information to determine the exposure of each individual amino acid side chain, thus providing a measure of how amenable that residue is for harboring a modification (Fig 1A). In contrast to experimentally derived structures, the in silico structures come with prediction errors and positional uncertainties, which turned out to be crucial for assessing amino acid exposure [17] (Fig 1B and 1C and S1 Fig). Our analyses confirmed that most PTMs are strongly enriched on exposed amino acids compared to residues that are buried within the protein fold (S2 Fig). This effect is even more evident when comparing IDRs and structured regions (Fig 2A) and for specific kinase phosphorylation motifs (Fig 3B and S3A Fig).

In contrast to the other analyzed PTMs, ubiquitinations were strikingly enriched on structured regions and on amino acids that are expected to be inaccessible (Fig 2A and S2 Fig). We showed that this effect is triggered by proteasome inhibition, supporting the idea that the ubiquitinations on structured regions are predominantly placed on misfolded proteins, tagged for

degradation. Thus, our toolkit can help distinguish ubiquitination associated with protein quality control from that mediating site-specific regulation. Interestingly, a majority of the proteins with ubiquitin modifications enriched in structured regions are DNA or RNA binders, many of which are known to be ubiquitinated in cellular stress response [37]. As an example, the RNA-binding protein G3BP2, which is essential for cytoplasmic stress granule formation, has all of its 9 ubiquitination sites in structured regions, 6 of which lie in the NTF2-like domain [23]. SET8 is ubiquitinated during DNA damage response, causing its degradation followed by chromatin rearrangements [38], and our analysis placed 9 of 10 ubiquitination sites in normally structured regions (adj. $p$-value = 0.01). Together, these results raise the possibility that ubiquitination of DNA and RNA binders and their direct regulators provides an effective regulatory mechanism for cellular stress responses. Coming back to G3BP2, the role of the NTF2-domain as oligomerization and USP10 binding site suggests that the ubiquitination of G3BP2 might impact the formation of protein assemblies. Recent progress in the prediction of protein complex structures will enable a systematic analysis of such effects in future studies [39,40]. Further investigations could combine the above analyses with information from linkage-specific ubiquitination studies (for instance, K48 versus K63) or even information about different ubiquitin side chain architectures to elucidate possible 3D topologies and associated functionalities.

As PTMs on properly folded proteins are expected to reside on exposed amino acid side chains, we reasoned that our 3D analysis could help to prioritize sites from experiments performed under less than physiological conditions. Here, we exemplified such a case by an in vitro kinase substrate screen [26]. While the screen allowed defining sequence preferences for kinase phosphorylation motifs, determining which sites could mediate bona fide regulation remained a challenge. Modified sites that are observed on inaccessible amino acid residues can be filtered out to reduce the target list to more physiologically relevant sites. Moreover, in our analysis, the motifs of best-retained sites were strikingly better defined than those based on the discarded ones (Fig 3D). We strongly suggest to employ such analyses in future in vitro PTM studies. Apart from phosphorylations, we expect similar benefits for the analysis of other types of PTMs or even for linear motifs involved in specific molecular binding events.

More generally, 3D analysis can highlight sites with a high potential to be functionally relevant. Our unbiased, global analysis revealed that regulatory phosphosites are strongly enriched in short disordered regions between large folded domains (Fig 4B). Many of these short IDRs correspond to annotated kinase activation loops, which are known to undergo structural rearrangements upon phosphorylation [30]. Our analysis systematically reveals such functionally highly relevant sequence regions on a proteome-wide scale, opening up multiple interesting routes for further investigations: First, short IDRs in regions without known functional relevance could specifically be investigated. Second, PTMs lying in or adjacent to such regions could be prioritized in the selection of candidate PTMs for biochemical follow-up studies, given their potential to cause structural rearrangements with functional consequences. We provide the community with a resource of such short IDRs in the human proteome and also with a set of promising PTM candidates from PhosphoSitePlus, which lie within or directly adjacent to these short IDRs. Beyond phosphorylation, these candidate PTMs contain hundreds of ubiquitinations as well as tens of sumoylations, acetylations, and methylations (Fig 4G). It would be exciting if future work in protein structure prediction enables the direct prediction of protein folds including PTMs.

Stepping back, the structural context of PTMs can not only be evaluated by means of spatial metrics, such as the exposure and folding state of individual amino acids, but also by using the three-dimensional fold directly. To this end, we evaluated PTM proximity in 3D space, revealing that not only phosphorylations, but also most other PTM types tend to cluster in protein

folds (Fig 5A). Beyond this colocalization of PTMs of the same type, we further showed that phosphorylations and lysine modifications preferentially occur in 3D proximity (Fig 5B and 5C), extending previous findings in linear sequence space [35]. Interestingly, lysine modifications, including ubiquitination, sumoylation, acetylation, and methylation, often compete for the same or directly neighboring residues, but they do not generally favor proximity (Fig 5C). A possible explanation for this effect could be the high reactivity of specific lysine residues, which will be an interesting follow-up to our study [41]. We further investigated PTM proximity on individual proteins in a proteome-wide fashion. This systematic analysis resulted in the identification of interesting proteins with 3D clusters of phosphorylations in specific protein pockets. For instance, in PDHA1, the relevant phosphorylation sites may all be important for enzyme activation and inactivation [36]. In AKR1B1, the relevant phosphorylations surround the known NADP binding site. We only observed functionally relevant PTM hotspots in protein pockets for phosphorylations here, but we expect that future improvements in the coverage of other PTM types will also enable the identification of 3D clusters for them. The current 3D proximity analysis builds directly on the structures predicted by AlphaFold. This comes with the limitation that IDRs and domains that are linked by IDRs are often not accurately predicted and their relative positions are less defined. In our analysis strategy, we account for this fact by removing long IDRs from the analysis and by considering the PAE for distance calculations. To further evaluate flexible protein regions in this type of analysis, ensemble structures could provide valuable insights, albeit being only available for a small subset of proteins to date [42]. Recent work extends the prediction of structures of individual proteins to the prediction of structures for entire protein complexes [39,40]. The integration of proteome-wide estimates of binding interfaces may make studies of PTM coregulation across proteins, such as the synergistic sumoylation of multiple proteins during DNA repair [43], feasible on a global scale.

In summary, our study, to our knowledge, is the first systematic assessment of the structural context of PTMs and how to leverage this information to gain novel biological insights and to augment proteomic studies. By assessing predicted protein structures for almost the entire human proteome, we here integrated PTM information at an unprecedented and comprehensive scale. Gratifyingly, for phosphorylation—the most frequently studied PTM type—our results confirm many previous findings about their structural preferences. They also reveal novel insights that expand to other modification types, exemplified by the observed accumulation of ubiquitination sites in structured regions. With the tools developed here, we have shown that comprehensive, in silico structural information can be used to prioritize physiological PTM sites from in vitro screens. Furthermore, they allowed us to select potentially regulatory sites by extracting PTMs in short IDRs; from phosphorylated kinase activation loops and extending to sequence regions with PTMs that are currently unexplored but likely to be of functional importance. Finally, based on the predicted structures, StructureMap identifies patterns of PTM proximity, revealing potential regulatory mechanisms that occur in 3D space. By providing the open-source Python packages StructureMap and AlphaMap, we equip the scientific community with a toolset to easily integrate information from AlphaFold-predicted structures into any kind of proteomics study and to visualize proteomics data on the predicted protein fold. We envision that this will enable scientists to directly assess observed PTMs in their structural context and to evaluate their physiological feasibility, to prioritize promising candidates for functional follow-up studies, and to find three-dimensional hotspots of PTM regulation. Future work could explore 3D motifs for enzyme substrate recognition as well as the integration of protein binding information. Beyond our own work presented here, we are convinced that the systematic integration of structural information with proteomics data will open up new avenues to (re)evaluate both old and new research questions.

## Methods

### Prediction of intrinsically disordered regions (IDRs) and benchmarking against IUPred2

To benchmark IDR prediction, we first downloaded ground truth datasets for both disordered and structured protein regions as defined by Beltrao and colleagues [17]. To avoid inconsistencies, we removed any amino acids that were labeled as both disordered and structured. We downloaded and formatted the AlphaFold structure predictions for 3,062 of the remaining 3,080 proteins. The other 18 proteins did not have a complete structure deposited in AlphaFold DB [13,14]. As a reference dataset for our benchmark, we obtained direct IDR estimates for all proteins from IUPred2A (https://iupred2a.elte.hu/) [20].

RSA values for all amino acids of the benchmarking proteins were calculated using DSSP from Bio.PDB (version 1.79) using default parameters [44]. Similar to Beltrao and colleagues, we applied a local smoothing of both the pLDDT and RSA values by averaging along the sequence with a window size of 5, 10, 15, 20, and 25 amino acids [17].

To evaluate the ability of our pPSE metric to predict IDRs, we calculated the pPSE values of all amino acids with different parameter settings. Given the goal of finding IDRs, we chose a constant angle term of 180˚. This corresponds to the full sphere exposure, which is independent of side chain orientation. For the distance, we screened over multiple parameters including 12 Å, 16 Å, 20 Å, 24 Å, and 28 Å. pPSE values were calculated both with and without considering the PAE. Similar to the pLDDT and RSA values, pPSE values were also smoothed by averaging along the sequence with a window size of 5, 10, 15, 20, or 25 amino acids.

The performance of IUPred2, pLDDT, RSA, and pPSE in predicting IDRs was evaluated based on the AUC and TPR at 5% FPR, when screening over different scoring thresholds. The best results in this analysis were obtained by the pPSE metric when considering PAEs and a distance threshold of 24 Å. Here, a smoothed pPSE $\leq$ 34.27 obtained a TPR of 85% at a FPR of 5%. IDR prediction for our proteome-wide analysis was therefore performed based on these parameters for pPSE calculation.

For evaluating the benchmark analysis, it is important to keep in mind that the annotations of structured regions and IDRs in the benchmark dataset were unfortunately not associated with a specified amino acid sequence, but only with sequence positions. While this should be correct for most proteins, there might be some differences between the original sequences for which the IDRs were predicted and the sequences used to predict structures by AlphaFold, thus potentially leading to slight inconsistencies.

All data and code for the IDR benchmark analysis are available at https://github.com/MannLabs/structuremap_analysis/blob/master/IDR_benchmark.ipynb.

### Definition of short IDRs

The goal of our "short IDR" detection strategy was to find flexible regions that are embedded into large folded domains. We defined short IDRs as amino acid stretches that we predicted to be IDRs and that comprise maximally 20 residues sandwiched between 2 flanking structured regions of at least 80 amino acids. Based on our strategy to identify IDRs by a sliding average of the pPSE exposure, some short IDRs only contain very few amino acids that reach the threshold to be classified as IDRs. To account for this effect, we introduced the extended regions of 5 amino acids on either side of short IDRs. We further expect the extension of short IDRs to be very beneficial in identifying potentially interesting PTMs, given that modifications on residues directly next to a flexible region could also affect its 3D conformation.

## pPSE parameters for estimating amino acid side chain exposure

The pPSEs for side chain expose estimation were calculated based on a distance threshold of 12 Å and an angle of 70˚. These values were chosen based on considerations about the average size of amino acids of approximately 3.5 Å and side chain flexibility around the direction of the beta carbon. Visualization of the resulting pPSE values showed expected patterns, as can be seen in Fig 1A (middle):

- Amino acids at the core of the protein fold have the highest pPSE values.

- Beta sheets have an alternating pattern of high and low exposure sites.

- Outwards-facing amino acids of alpha-helices have a higher exposure than inwards-facing sites.

- Amino acids in badly predicted sequence regions (e.g., the tailing sequence) have a pPSE value close to 0.

To categorize amino acids into low versus high exposure, we selected a pPSE threshold of 5. This means that amino acids with a pPSE $\leq 5$ were considered to have a high exposure. S1B Fig illustrates the distribution of pPSE values across all amino acids in structured protein regions (non-IDRs).

Due to missing ground truth data, we could not perform a comprehensive parameter screen and robustness test. However, based on considerations of the molecular distances involved and on exemplary visualizations, we expect meaningful exposure metrics for parameters ranging from approximately 10 to 25 Å distance and an angle between approximately 45 and 90˚. Interpretability then depends on different pPSE thresholds. In future studies, it may be useful to adjust parameters depending on the research question and possibly the type and size of modification types that are being assessed.

## PTM enrichment analyses

All enrichment analyses were performed using the two-sided Fischer's exact test available in scipy.stats version 1.7.1 [45]. *P* values were subsequently adjusted for multiple testing [46] using statsmodels.stats.multitest.multipletests version 0.13.0 [47].

## Motif analysis with PSSMMatch

The kinase-specific phosphosites were extracted from S2 Table from Sugiyama and colleagues [26]. UniProt IDs were matched to UniProt accessions, and the sequence windows corresponding to ±6 amino acids around each phosphosite were extracted from the current human protein fasta file (version: 2021_03, downloaded on 02.08.2021). Any sites not matching the expected phosphoacceptor residue were removed from the analysis. For each selected kinase, targeted sequence windows were filtered as follows:

1. All reported sequence windows with a phosphosite pPSE $\leq 5$. The number of resulting sequence windows is defined as $N_{exposed}$.

2. All reported sequence windows with a phosphosite pPSE $> 5$. The number of resulting sequence windows is defined as $N_{not\text{-}exposed}$. Usually, $N_{exposed} > N_{not\text{-}exposed}$.

3. A randomly selected subset of size $N_{not\text{-}exposed}$ of the reported sequence windows with a phosphosite pPSE $\leq 5$.

Motif analysis was subsequently performed using the PSSMSearch tool for each of the 3 sets of sequence windows by selecting "log odds" as scoring method [27,28].

## Gene ontology enrichment analysis with DAVID

All gene ontology enrichment analyses presented herein were performed on the DAVID platform version 6.8 [48,49].

For the enrichment analysis of proteins with short IDRs, we compared the 2,454 proteins with short IDRs against the background of all 20,053 human proteins with available structural information. Direct Gene Ontology terms of the "Molecular function" category were considered. Filtering was set to an FDR threshold of 0.01, a minimum fold enrichment of 2 and a minimum count of 10.

## PTM import from PhosphoSitePlus

Lists of PTMs were downloaded from PhosphoSitePlus on 01.08.2021 [21]. PTM types include phosphorylations (p), ubiquitinations (ub), sumoylations (sm), acetylations (ac), methylations (m), and the glycosylations O-GalNAc (ga) and O-GlcNAc (gl). Additionally, the set of regulatory PTMs was downloaded and filtered for the modifications mentioned above. For each PTM type, sites were filtered for a selected set of the most common acceptor residues as follows:

- p: S, T, Y

- ub, sm and ac: K

- m: K, R (Note: all types of methylations were grouped together)

- ga, gl: S, T

In rare cases where regulatory sites were reported for a specific PTM type, but no matching entry was found in the according dedicated PTM list, these sites were added there.

The data processing of PhosphoSitePlus data is available at https://github.com/MannLabs/structuremap_analysis/blob/master/ptm_data_import/import_phosphositeplus.ipynb.

## Global PTM proximity analysis

In our global PTM proximity analysis, we investigated whether PTM acceptors near a modified amino acid residue are more frequently observed to also be modified compared to more distant residues or random expectation, a concept previously introduced by Beltrao and colleagues [35]. Here, we extended the concept to evaluate PTM proximity in 3D space and to assess both individual PTM types and PTM co-occurrence. Starting from a set of observed modifications, e.g., phosphorylations, we calculate the 3D distance to all observed modifications of either the same type, e.g., also phosphorylations, or of a different type, e.g., ubiquitinations. Importantly, we consider the positional uncertainty of AlphaFold predictions by adding the PAE to each distance. We also generate 5 random permutations, where the same number of modifications are distributed randomly across all possible acceptor residues. Finally, we select distance bins and count the number of modified residues in each of these bins for both the real observations and the randomized background. To ensure that disordered regions do not bias the results, only structured regions and short IDRs were considered in this analysis (i.e., we removed all IDRs stretching more than 20 amino acids).

For PTM self-proximity, we started distance bins at 1 Å ranging up to a maximum of 35 Å in step sizes of 5 Å. For PTM colocalization analysis, we started distance bins already at 0 Å to

enable the evaluation of competition for the same site, especially between different lysine modifications such as ubiquitination, sumoylation, and acetylation.

## Per protein PTM cluster analysis

To find proteins with a significant colocalization of PTMs, we calculated all pairwise distances between the alpha carbons of modified acceptor residues. The matching PAE provided by AlphaFold was added to each calculated distance to account for positional uncertainties. The average distance was subsequently compared to the average distances of 10,000 random permutations among the modified acceptor residues, thus resulting in empirical *p*-values. These *p*-values were adjusted for multiple testing [46] using statsmodels.stats.multitest.multipletests version 0.13.0 [47].

## Protein structure visualization with AlphaMap

The 3D structure visualization in AlphaMap was implemented by integrating a Mol* Viewer [50]. Code was adapted from https://github.com/molstar/pdbe-molstar.

## Supporting information

**S1 Fig. Estimation of amino acid side chain exposure and IDRs. (A)** Visualization of the strategy to calculate the pPSE. **(B)** Distribution of pPSE values across all amino acids in structured protein regions (non-IDRs). **(C)** Parameter screen to evaluate the ability of different metrics to predict IDRs based on the TPR at a 5% FPR. Source data are available at https://github.com/MannLabs/structuremap_analysis/blob/master/data/alphafold_data/pPSE_bincount_df.csv. **(D)** Parameter screen to evaluate the ability of different metrics to predict IDRs based on the AUC. The numbers in square brackets behind each metric indicate the smoothing windows that were used. Source data for (C) and (D) are available at https://github.com/MannLabs/structuremap_analysis/blob/master/data/disorder_data/IDR_ROC_curve.csv. AUC, area under the curve; FPR, false positive rate; IDR, intrinsically disordered region; pPSE, prediction-aware part-sphere exposure; TPR, true positive rate.
(EPS)

**S2 Fig. Enrichment analysis of PTMs in amino acids with high versus low side chain exposure. (A)** Enrichment of different PTMs annotated in the PhosphoSitePlus database in amino acids with side chains of high side chain exposure within structured regions. PTMs are abbreviated as follows: phosphorylations (p), ubiquitinations (ub), sumoylations (sm), acetylations (ac), methylations (m), and the glycosylations O-GlcNAc (gl) and O-GalNAc (ga). **(B)** Enrichment of ubiquitinated lysines annotated in PhosphoSitePlus versus ubiquitinations detected in a dataset treated with proteasome inhibitor or untreated. Source data for (A) and (B) are available at https://github.com/MannLabs/structuremap_analysis/blob/master/data/ptm_enrichment/ptm_enrichment_high_acc_5.tsv. PTM, posttranslational modification.
(EPS)

**S3 Fig. Exploiting the 3D context of kinase phosphorylation motifs. (A)** Enrichment of phosphorylations in kinase motifs with amino acids of high side chain exposure compared to all possible kinase motif occurrences in structured regions. Source data are available at https://github.com/MannLabs/structuremap_analysis/blob/master/data/ptm_enrichment/enrichment_p_inMotif_inHighAcc.tsv. **(B)** Sequence logos for different kinases based on a random subset of high-exposure sites, comprising the same number of sites as compared to the low-exposure set. The PSSMSearch tool (Krystkowiak and colleagues [27]) was used with a

log odds scoring method (O'Shea and colleagues [28]).
(EPS)

**S4 Fig. PTM proximity analysis in 3D. (A)** The fraction of modified PTM acceptor residues is shown as a function of the 3D distance to a given modified amino acid in Å. Observed values (indicated in red when statistically significant and colored in salmon otherwise) are compared to the mean of 5 random samples including the same number of modified PTM sites (gray). Error bars indicate 1 standard deviation. The x-axes are divided in distance bins ranging from each previous bin to the indicated cutoff in Å. Source data are available at https://github.com/ MannLabs/structuremap_analysis/blob/master/data/proximity_analysis/Fraction_of_ modified_acceptor_residues_self_noIDRs.csv. **(B)** The fraction of modified phospho-acceptor residues is shown as function of the 3D distance to a given modified amino acid in Å. Source data are available at https://github.com/MannLabs/structuremap_analysis/blob/master/data/ proximity_analysis/Fraction_of_modified_acceptor_residues_p_colocalization_noIDRs.csv. **(C)** The fraction of ubiquitinated lysines is shown as function of the 3D distance to a given modified amino acid in Å. The smallest bin shows competition for the same central lysine residue. Source data are available at https://github.com/MannLabs/structuremap_analysis/blob/ master/data/proximity_analysis/Fraction_of_modified_acceptor_residues_ub_colocalization_ noIDRs.csv. PTM, posttranslational modification.
(EPS)

**S1 Dataset. PTMs in short intrinsically disordered regions.** PTM, posttranslational modification.
(XLS)

**S2 Dataset. Predicted short IDRs.** IDR, intrinsically disordered region.
(XLS)

**S3 Dataset. Proteins with significant phosphosite clusters.**
(XLS)

## Acknowledgments

We thank Julia P. Schessner for valuable discussions about the implementation of the pPSE exposure metric. We also thank J. Rajan Prabu for sharing his knowledge in structural bioinformatics.

## Author Contributions

**Conceptualization:** Isabell Bludau, Matthias Mann.

**Data curation:** Isabell Bludau, Fynn M. Hansen, Maria C. Tanzer, Ozge Karayel.

**Formal analysis:** Isabell Bludau.

**Funding acquisition:** Isabell Bludau, Matthias Mann.

**Investigation:** Isabell Bludau, Fynn M. Hansen, Maria C. Tanzer, Ozge Karayel, Brenda A. Schulman, Matthias Mann.

**Methodology:** Isabell Bludau.

**Project administration:** Matthias Mann.

**Resources:** Isabell Bludau, Maximilian T. Strauss.

**Software:** Isabell Bludau, Sander Willems, Wen-Feng Zeng, Maximilian T. Strauss.

**Supervision:** Matthias Mann.

**Validation:** Isabell Bludau.

**Visualization:** Isabell Bludau, Wen-Feng Zeng.

**Writing – original draft:** Isabell Bludau, Matthias Mann.

**Writing – review & editing:** Isabell Bludau, Sander Willems, Fynn M. Hansen, Maria C. Tanzer, Ozge Karayel, Brenda A. Schulman, Matthias Mann.

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
