## [Editor Report · Decision Letter 0]

8 Mar 2022

Dear Dr Mann, 

Thank you for submitting your manuscript entitled "The structural context of PTMs at a proteome wide scale" for consideration as a Research Article by PLOS Biology.

Your manuscript has now been evaluated by the PLOS Biology editorial staff and I am writing to let you know that we would like to send your submission out for external peer review. We would like to consider your submission as a 'Methods and Resources' article (more Resource than Method), so we ask that you please tick this as the article type upon resubmission (see below). 

Before we can send your manuscript to reviewers, we need you to complete your submission by providing the metadata that is required for full assessment. To this end, please login to Editorial Manager where you will find the paper in the 'Submissions Needing Revisions' folder on your homepage. Please click 'Revise Submission' from the Action Links and complete all additional questions in the submission questionnaire.

Once your full submission is complete, your paper will undergo a series of checks in preparation for peer review. Once your manuscript has passed the checks it will be sent out for review. Given the competition in this topic area, we will do all we can to ensure a quick peer review process. To provide the metadata for your submission, please Login to Editorial Manager (https://www.editorialmanager.com/pbiology) within two working days, i.e. by Mar 10 2022 11:59PM.

If your manuscript has been previously reviewed at another journal, PLOS Biology is willing to work with those reviews in order to avoid re-starting the process. Submission of the previous reviews is entirely optional and our ability to use them effectively will depend on the willingness of the previous journal to confirm the content of the reports and share the reviewer identities. Please note that we reserve the right to invite additional reviewers if we consider that additional/independent reviewers are needed, although we aim to avoid this as far as possible. In our experience, working with previous reviews does save time. 

If you would like to send previous reviewer reports to us, please email me at rhodge@plos.org to let me know, including the name of the previous journal and the manuscript ID the study was given, as well as attaching a point-by-point response to reviewers that details how you have or plan to address the reviewers' concerns. 

Given the disruptions resulting from the ongoing COVID-19 pandemic, please expect some delays in the editorial process. We apologise in advance for any inconvenience caused and will do our best to minimize impact as far as possible.

Kind regards,

Richard

Richard Hodge, PhD

Associate Editor, PLOS Biology

rhodge@plos.org

PLOS

---

## [Decision Letter · Decision Letter 1]

29 Mar 2022

Dear Matthias,

Thank you for your patience while we peer-reviewed your manuscript "The structural context of PTMs at a proteome wide scale" as a Methods and Resources study at PLOS Biology. Your manuscript has been evaluated by the PLOS Biology editors, an Academic Editor with relevant expertise, and by several independent reviewers. I have taken over the handling of tis submission during Richard's absence from the office this week, to prevent any unnecessary loss of time.

As you will see from the reviews at the end of this email, all of the reviewers are very supportive of the work. Reviewers 2 and 3 have identified themselves as Jon Agirre and Julian Langer, respectively. Reviewer 1 has raised more critical concerns despite the overall support, and upon discussion with our Academic Editor we consider the request to look at deposited ensembles in the PEDB to be a good one, but if there are not enough submissions for the analyses, then the concerns could be addressed by textual changes by clarifying the limitations/parameters of the 3D hotspot model.

In light of the reviews, we are pleased to offer you the opportunity to address all of the concerns raised by the reviewers in a revised version that we anticipate should not take you very long. We will then assess your revised manuscript and your response to the reviewers' comments with our Academic Editor and may consult (as subset of) the reviewers again. We expect to receive your revised manuscript within 1 month, but please let us know if you would need longer.

In addition, please ensure that your revision also addresses the following reporting and/or formatting concerns (this will be needed for eventual acceptance, so doing this now will speed up the process):

- it is PLOS policy that, as a condition of publication, you make available all data used to draw the conclusions arrived at in your manuscript (http://journals.plos.org/plosbiology/s/data-availability). The summary numerical values underlying the graphs all of the main and supplementary figures should be provided, either in a source data file within the manuscript's supplementary information or, preferably, in a public repository. All figure legends must then include information on where the underlying source data for the figure can be found (if this is a repository, the name of the specific file needs to be indicated, rather than e.g. a folder or directory). Please note that this does not refer to the raw data analysed, which we also encourage to provide, but the summary numerical values that are directly related to the figures shown. 

- would you be able to add an explanation for the grey column in fig. 2B in the figure legend? It strikes as a bit odd that the other columns are based on specific references, but then 677 papers are shown together. I realise that this explanation may be elsewhere in the manuscript, but a lot of readers focus on the figures and legends, so it should also be there.

- the references are not in PLOS Biology format; please see https://journals.plos.org/plosbiology/s/submission-guidelines#loc-formatting-references

- the author contributions, and conflict of interest statements will be taken from the information that you enter in our submission system for publication. Please delete these sections from the manuscript, but when you submit your revision, ensure the information in the submission system is complete and accurate

- likewise, in the financial disclosure information (which is also on our electronic submission system), please ensure that all sources of funding are identified with the relevant ID/number

- Matthias, we have fixed the duplicate account issue, and so you should now be able to link your ORCiD to the submission while submitting the revised version without a problem (we cannot do this bit on your behalf) - do let me know if this is not solved, but I am told it is

Please email us (plosbiology@plos.org) if you have any questions or concerns about the revision. At this stage, your manuscript remains formally under active consideration at our journal; please notify us by email if you do not intend to submit a revision so that we may end consideration of the manuscript at PLOS Biology.

**IMPORTANT - SUBMITTING YOUR REVISION**

*Resubmission Checklist*

*Published Peer Review*

*PLOS Data Policy*

With best wishes,

Nonia

Nonia Pariente, PhD

Editor in Chief

PLOS Biology

on behalf of

Richard

Richard Hodge

Associate Editor

PLOS Biology

rhodge@plos.org

REVIEWS:

Reviewer #1: 

The authors report results from their analysis of sites for PTMs that are based on the use of predicted protein structures. The latter were derived from AlphaFold. The question, as it comes through in my reading of the MS, is the extent to which predicted structures hold insights regarding the different PTM patterns. For this, the authors develop metrics for different types of PTMs, based largely on accessibility of the different sites for different PTMs, and evaluate their results based on extant knowledge bases. The results corroborate what has been established regarding phosphorylation. The results do, however, offer interesting and novel insights regarding what we think we know about ubiquitination and acetylation. 

Some of the findings, especially those pertaining to Ser / Thr phosphorylation are not really surprising or novel. This has a lot to do with the extents to which these modifications have been over-studied. It would help to have a crisp summary of the novel findings that emerge. For example, it is well established (sorry for using this trope) that phosphosites are enriched within IDRs. What more do we learn from the current analysis? The answer to this question was not entirely clear. One helpful insight was the comparative analysis of phosphosites gleaned from the use of structure predictions vs. the analysis of Sugiyama et al. This is a helpful comparison inasmuch as it endorses the range of studies that have highlighted the importance of IDRs as harboring phosphosites. 

Among the interesting findings are observations that ubiquitination is preferentially targeted to sites with low sidechain exposure scores. This has important and interesting implications for the impact of unfolding on ubiquitination. In this context, the G3BP2 result is also very interesting. The sites of ubiquitination are localized (mostly) to the NTF2L dimerization / oligomerization domain. This suggests a linkage between the monomer - dimer / oligomer equilibrium and ubiquitination. Accordingly, in addition to a linkage with folding - unfolding equilibria, it follows that ubiquitination might preferentially target folded domains that form higher order oligomers. It would be very interesting to investigate this issue and put the findings on a quantitative footing. 

While the smoothed pLDDT scores are indeed useful for identifying IDRs, using the specific AlphaFold structure for the IDR can be problematic. Please clarify if this is indeed the case for the calculation of pPSE. If this is the case, then how does one test the validity of the score, seeing as this will be useful for comparative assessments. The problem here is the reduction of an ensemble to a single, essentially randomly drawn structure for the IDR. Comparisons made to date between the static structures and data from in vitro biophysical characterizations of IDRs show that the static structure of an IDR within AlphaFold predictions bears no resemblance to any of the ensembles that have been characterized using measurements integrated with computations. One, hopefully constructive suggestion would be to quantify the extent to which pPSE scores computed using the current method agree with scores computed using ensembles that have been deposited in the PEDB. 

Please define STY sites. Does this simply mean sites containing Ser, Thr, and Tyr residues? Also, from a semantic and conceptual standpoint, classifying regions as being unstructured is misleading. They are disordered, which has a very different connotation compared to the term unstructured. Please consider not using the latter term because disordered ≠ unstructured. 

The aspect of the study that is concerning is the extraction of 3D hotspots. This is an intriguing hypothesis, although the argument and approach are concerning. Phosphosites are enriched in IDRs. The authors establish this quite clearly. To then remove the IDRs and propose that the phosphosites occur in 3D clusters places a high emphasis on the structurally resolved domains. In effect, as I understand it (and I could be wrong), the analysis suggests that because a region that is the target of PTMs tethers domains that are spatially proximal, it follows that the sites of PTMs form clusters in 3 dimensions. This does not have to follow, although one can see some physical basis for this hypothesis. It is an intriguing hypothesis that merits close scrutiny, perhaps using simulations or by leveraging calculated ensembles deposited in the PEDB. Similar concerns apply to the calculation of distances between modified sites. What is not clear is if the authors are relying on the static structures provided by AlphaFold. This is highly problematic if true. 

Reviewer #2 (Jon Agirre): 

This article by Bludau and collaborators present an interesting study of AlphaFold-powered structural context for post-translational modifications in proteins at the proteome level. This work employs a similar methodology to what we are doing in my lab with protein glycosylation (for a sneak peek, see Bagdonas, H., Fogarty, C. A., Fadda, E., & Agirre, J., 2021, The case for post-predictional modifications in the AlphaFold Protein Structure Database. Nature Structural & Molecular Biology, 28(11), 869-870), so on the plus side I fully agree with it, and on the negative side I as a reviewer cannot bring an entirely different point of view to the discussion. I believe the decision to use *just* AlphaFold models is justified, as most models will be of at least 'OK' accuracy, and it simplifies the procedure greatly. I think overall there is a good grasp of the technicalities of AlphaFold here, and the limitations are discussed adequately. 

I have got a couple of suggestions:

- The lettering on the figures is way too small in many cases (Figs. 4 & 5 for instance). 

- Perhaps I missed it, but I could not find a section on reproducibility. It's great that the source code is available on GitHub, but perhaps the data used for the graphs could also be made available as a zenodo download? 

Reviewer #3 (Julian Langer): 

In their manuscript, Bludau et al. present an elegant approach to incorporate information on a variety of PTMs with the prediction of structural information on proteins harboring modification sites. Using previous knowledge on functionally-relevant modification sites (including but not limited to phosphorylation) in combination with deep-learning assisted structure-prediction by alphaFold2, the authors propose strategies to assist in discovery of PTM sites with functional potential, i.e. PTMs within or in proximity to short disordered regions. 

MS-based methods and the ability to detect PTMs has been constantly improving over the last years. However, bioinformatics tools and approaches to systematically analyze and make sense of the plethora of quantified modification sites has not caught up with comparable pace. Hence, the analysis pipeline presented and made accessible for readers in this manuscript is of broad interest for a general audience in molecular, structural and systems biology and thus suitable for publication in PLOS Biology.

We appreciate the thoroughness of the authors' investigation and have only minor comments and suggestions to improve the readability of the manuscript, and two minor biological questions the authors could comment on.

Minor comments:

* Page 4: We think it would be beneficial if the authors would explain the selection of parameters for pPSE calculation in greater detail: How did they decide on 12 Å and an angle of 70°? How robust are the findings with respect to slight changes of these settings?

* Typo: "proteome wide" in title, "proteome-wide" in text

* Figure 2: It would be helpful, if the authors provided some more details in the axis-labels, i.e. 'p_reg' = regulated phospho-events.

* Page 6: There seems to be a typo '… ubiquitination sites from (written: form) proteasome-inhibitor-treated …'

* Page 9: The authors expand on sequence motif analysis by testing the effect of 3D-exposure of the phosphorylation site, they further exploit the Stukalov et al. data for this, but did not mentioned it in the very same paragraph (but before and figure legend). We believe it can be helpful to repeat the source of the data in the first sentence of this paragraph as it prevents the reader to get lost in the text.

* Page 10: It would improve readability if the authors added some more description in the figure legend for the motif, i.e. mentioning the selection with respect to surface expose (left) / (right). 

* Page 13, Figure 4G: Labeling of bar graph on the left (x-axis) could be more detailed with respect to "p_functional_0" and "p_functional_5".

* Page 11+12: Functional score of phospho-sites. Is this derived from Ochoa et al.. It is not clearly mentioned where the authors extracted that piece of information

Minor questions:

* Page 6: Given their similarity, could the authors comment on the observed discrepancy between sumoylation and ubiquitination in Figure 2A, esp. for sites lacking functional annotation? 

* Page 7: The analysis of ubiquitination in structured vs. unstructured region is very interesting. It would be interesting to investigate proteomics datasets of linkage-specific ubiquitin-enrichments (K48 vs. K63) to look for potential differences in these two modification with respect to the site of modification.

---

## [Editor Report · Decision Letter 2]

19 Apr 2022

Dear Dr Mann,

On behalf of my colleagues and the Academic Editor, Kylie Walters, I am pleased to say that we can in principle accept your Methods and Resources article "The structural context of post-translational modifications at a proteome-wide scale" for publication in PLOS Biology, provided you address any remaining formatting and reporting issues. These will be detailed in an email that will follow this letter and that you will usually receive within 2-3 business days, during which time no action is required from you. Please note that we will not be able to formally accept your manuscript and schedule it for publication until you have completed any requested changes.

We have taken the liberty of changing the title on your behalf to ‘The structural context of post-translational modifications at a proteome-wide scale’, just to spell out the PTM acronym. If you would prefer to change it back, then you will still have the chance to do so during the production process.

We have also noted your request to shorten the links to the GitHub repository in the figure legends. We have let production know about this and I will personally check, once the proofs have been completed, to ensure that this has been done. 

PRESS

Sincerely, 

Richard

Richard Hodge, PhD 

Associate Editor, PLOS Biology

rhodge@plos.org

PLOS
